# Differentiating lightning in winter and summer with characteristics of wind field and mass field

Deborah Morgenstern[1,2], Isabell Stucke[1,2], Thorsten Simon[2], Georg J. Mayr[1], and Achim Zeileis[2]

[1]Department of Atmospheric and Cryospheric Sciences (ACINN), University of Innsbruck, Austria
[2]Department of Statistics, University of Innsbruck, Austria

**Correspondence:** Deborah Morgenstern (deborah.morgenstern@uibk.ac.at)

**Abstract.** Lightning in winter (December, January, February, DJF) is rare compared to lightning in summer (June, July, August, JJA) in central Europe north of the Alps. The conventional explanation attributes the scarcity of lightning in winter to seasonally low values of variables that create favorable conditions in summer. Here we systematically examine whether different meteorological processes are at play in winter. We use cluster analysis and principal component analysis and find physically meaningful groups in ERA5 atmospheric reanalysis data and lightning data for northern Germany. Two thunderstorm types emerged: wind-field thunderstorms and CAPE-thunderstorms. Wind-field thunderstorms are characterized by increased wind speeds, high cloud shear, large dissipation of kinetic energy in the boundary layer, and moderate temperatures. Clouds are close to the ground and a relatively large fraction of the clouds is warmer than $-10\,°C$. CAPE-thunderstorms are characterized by increased convective available potential energy (CAPE), the presence of convective inhibition (CIN), high temperatures, and accompanying large amounts of water vapor. Large amounts of cloud-physics variables related to charge separation such as ice particles or cloud base height further differentiate both wind-field thunderstorms and CAPE-thunderstorms. Lightning in winter originates in wind-field thunderstorms whereas lightning in summer originates mostly in CAPE-thunderstorms and only a small fraction in wind-field thunderstorms. Consequently, typical weather situations of wind-field thunderstorms in the study area in northern Germany are strong westerlies with embedded cyclones. For CAPE-thunderstorms, the area is typically on the anticyclonic side of a southwesterly jet.

**Keywords:** ERA5, cold-season thunderstorm, $k$-means clustering, PCA, winter lightning.

## 1 Introduction

Mid-latitude thunderstorms are much rarer in winter than in summer and produce less than $3\,\%$ of the total lightning activity in Europe (Wapler, 2013; Poelman et al., 2016). Yet the transported electrical charges are often higher in winter and thus the damage potential is also higher. The conventional explanation for the paucity of winter lightning is the paucity of favorable conditions for strong convection, which lead to thunderstorms in summer. The required large values of convective available potential energy (CAPE), copious amounts of near-surface water vapor and the presence of a vertical instability (Doswell III, 1987) are normally absent in winter.

The electrical characteristics of lightning in winter differ from summer, e.g., in flash duration, direction and sign of charge transfer, strength of the electric current, and the lightning electric field waveform (e.g., Brook et al., 1982; Goto and Narita, 1995; Rakov and Uman, 2003; Rakov, 2003; Diendorfer et al., 2009; Ishii and Saito, 2009; Wang and Takagi, 2012; Yoshida et al., 2018; Wu et al., 2021). Larger transported charges and more frequent initiation of lightning from tall (human-made) structures in winter elevate the damage potential. This has become a major concern as a consequence of the proliferation of the installation of tall wind turbines in the push towards renewable energy sources. For example, Matsui et al. (2020) show that wind turbine lightning accidents in Japan in winter are 47 times more frequent and also more severe than in summer.

The difference in electrical characteristics warrants to challenge conventional wisdom for the paucity of winter thunderstorms and investigate whether it is not meteorological settings different from summertime ones that lead to them. One therefore will need to look first at the processes that create lightning. While no unified theory exists that explains the build-up of the charge separation that lightning eventually neutralizes, the non-inductive mechanism is the most widely accepted one (Saunders, 2008; Williams, 2018). It states that charge is transferred during the collision of different cloud particles often present in the vicinity of the $-10\,°C$ isotherm. The differently charged particles get separated based on their size through differential terminal velocities (Cotton et al., 2011) and form various charge regions within the cloud. Lightning is initiated in the strong electric field between two charge regions (e.g., Salvador et al., 2021). In summertime, the release of CAPE leads to strong updrafts that are needed to produce graupel – relatively large and heavy hydrometeors – and to move ice crystals far aloft that have acquired opposite polarity through their collision with graupel (Williams, 2018). In wintertime, it is rather the collision between snowflakes and ice crystals and their subsequent separation along a slanted path that is thought to be responsible for the charge separation (Williams, 2018). Differential terminal velocities with strong vertical shear of the horizontal wind cause the particle paths to become slanted and separation distances to be large despite relatively weak vertical motions and charging rates. Lightning in winter occurs with clouds that are shallow but wide, a charge region that is close to the ground, and lightning discharges that propagate long distances within the cloud resulting in large charge transfers (Yoshida et al., 2018).

The goal of this paper is to take a step back from the obvious seasonality of lightning frequency (Vogel et al., 2016; Matsui et al., 2020) and apply a data-driven approach to elucidate whether the occurrence of lightning can be tied to different dominant meteorological processes. It is important to remember that lightning is not necessarily a synonym to "strong convection" since processes other than strong vertical motions might lead to charge separation and the electrification of clouds. If thunderstorm types are differentiated by processes instead of seasons, more insights can be gained and a contradiction arising from a seasonal classification can be resolved, for example, that of the annual lightning maximum in fall in the northern Mediterranean compared to central Europe where lightning peaks in summer (Taszarek et al., 2019). To clearly make the distinction between processes and a mere seasonality of favorable thunderstorm conditions, we focus on winter and summer seasons only at a fairly small and flat study region to avoid having topography as an additional forcing mechanism and to have homogeneous lightning conditions with a uniform annual lightning cycle over the entire domain. Results for the transition seasons are given in the supplements to this paper.

Our data-driven approach uses many atmospheric variables of possible relevance for thunderstorms associated with the wind field, mass (temperature) field, moisture field, surface exchange processes and cloud (micro-) physics from a meteorological

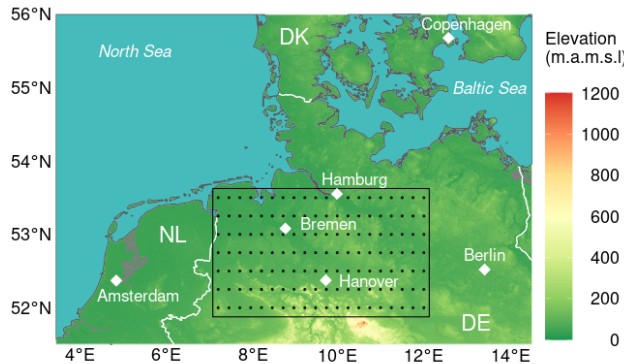

**Figure 1.** Study region in northern Germany (black rectangle). Coordinates: $52.00^\circ$ N, $7.25^\circ$ E (SW corner) and $53.50^\circ$ N, $12.00^\circ$ E (NE corner). Area: $53,295\,\mathrm{km}^2$. Dots show the centers of the ERA5 $0.25^\circ$ latitude/longitude grid. Elevation is mostly below $100\,\mathrm{m}$ amsl. Data: TanDEM-X (Rizzoli et al., 2017; Wessel et al., 2018).

reanalysis (ERA5) and lightning observations (both described in Sect. 2). The statistical methods establishing links between
meteorological data and lightning are described in Sect. 3. Sect. 4 presents the results and Sect. 6 discusses and summarizes the findings.

## 2   Data

The study area was chosen to be in the mid-latitudes, to be covered by a lightning location system with high detection efficiency, and to be topographically fairly uniform. A region in northern Germany shown in Fig. 1 fulfills these criteria. It includes some
small hills but the elevation is mostly some decameters above mean sea level.

The study period is 2010–2019, a period for which lightning detection efficiency in the study region is mostly unaffected by changes to hardware and software of several lightning locations systems (LLS) collaborating as EUCLID (European Cooperation for Lightning Detection). We use only cloud-to-ground lightning flashes since these are responsible for most damages. An additional amplitude filter is applied to exclude flashes with weak peak currents between $-5\,\mathrm{kA}$ and $15\,\mathrm{kA}$ resulting in a
detection efficiency of more than $96\,\%$ (Schulz et al., 2016; Poelman et al., 2016). From 2010–2019 EUCLID recorded 203,124 such flashes in the study region in summer (June, July, August, JJA) but only 2,830 in winter (December, January, February, DJF; $1.4\,\%$ of the flashes in summer).

Consistent atmospheric data come from ERA5, the fifth generation global reanalysis of the European Centre for Medium-Range Weather Forecasts (ECMWF; Hersbach et al., 2020). We use assimilated data at the surface level and data on the lowest
(of 137) vertical levels (covering the troposphere) and many additional variables derived from these data (see Sect. 3). Horizontally, the data are available on a $0.25^\circ$ latitude/longitude grid, temporally every hour, yielding a "cell-hour" as the smallest space-time unit. Only a small fraction of cell-hours have at least one flash in JJA (27,305; $0.883\,\%$) and even 17 times less in DJF (1,576; $0.052\,\%$).

## 3   Methods

To clearly isolate the effects of seasonality, only the two extreme seasons winter and summer are chosen and a methodological approach is selected that can properly handle the vastly different lightning frequencies in these two seasons. The same methods have been applied to the transitional seasons, for which results are given in the supplement.

To understand the atmospheric conditions under which lightning occurs (or not) we process the available EUCLID lightning observations and ERA5 atmospheric variables in the following way. First, equally-sized samples from four scenarios of lightning observations are formed: Lightning in winter, no lightning in winter, lightning in summer, and no lightning in summer, each following the diurnal cycle of lightning in the respective season (Sect. 3.1). To capture the atmospheric conditions at the time and place of these EUCLID observations, we select and derive 35 ERA5 variables at the respective grid cells (Sect. 3.2). Using only these 35 ERA5 variables a $k$-means cluster analysis with $k = 5$ clusters is carried out to determine groups of "typical" atmospheric conditions. To facilitate the interpretation of the 35 variables in the five clusters, the variables are visualized by the first two components of a principal component analysis (Sect. 3.3). Matching the membership for the five atmospheric condition clusters with the corresponding four lightning scenarios reveals how the atmospheric conditions vary between winter and summer with and without lightning. Finally, clusterwise weather maps are produced to get an overview of the governing weather patterns in each cluster and hence a good description of the differences between lightning in winter and in summer.

### 3.1   Composition and stratification of data

The EUCLID observations are aggregated to the spatio-temporal grid of ERA5. A cell-hour is considered as a lightning cell if at least one flash occurred within the cell in the hour after the ERA5 valid time. Otherwise the cell-hour is considered as non-lightning.

For best results of the clustering and principal component analysis, each of the four lightning scenarios considered should be represented equally in the data. Therefore, we use *all* cell-hours from the least frequent scenario (lightning in winter) along with samples of the same size from the other three scenarios. This sampling is done conditional on the diurnal cycle for lightning in the respective season, known as "stratified sampling" in statistical literature. All sampling is performed without replacement and on the basis of cell-hours.

Since the smallest scenario, lightning in winter, consists of 1,576 cell-hours, the whole data set with four scenarios contains 6,304 cell-hours. Finally, to ensure that the results obtained are not driven by spurious artifacts from the sampling, we have considered 50 replications of the sampling procedure. As all of these lead to qualitatively identical results, we only report the results from one representative set of samples. Each sample is drawn from the whole 10 years of data so that single anomalous seasons do not have a proportionally large influence. The similarity of the 50 samples gives further confidence in the robustness of our results. The representative data set is provided as an online supplement along with this paper.

## 3.2 Preprocessing and selection of ERA5 variables

To enhance the set of ERA5 single-level variables, we add information from the vertical profiles available in the model level data by deriving additional single-level variables from them. These derived variables aim at portraying physical lightning processes and cover isotherm heights, cloud size, wind shear within and below the cloud, and the maximum vertical velocity. Further, we compute sums of cloud particles between specific isotherms, for instance, cloud ice water content between the $-20\,°C$ and $-40\,°C$ isotherms. Table 1 presents all variables used in this study, the derived variables are marked by an asterisk.

An extended version of this table is provided in the supplements of this paper.

The 35 variables presented in Table 1 are selected subjectively from the extended ERA5 data set based on our own meteorological expertise, results in the literature, and an explorative analysis of the data. This explorative analysis worked out variables that show a distinct distribution for the four scenarios and we kept only variables that are not strongly correlated to other selected variables. The chosen atmospheric variables contribute to the formation and ultimately to the separation of

electric charges needed for lightning to occur. Each variable is associated with a physical-based category (Table 1):

- Mass field: Variables related to temperature and pressure such as CAPE and the altitude of specific isotherms.

- Wind field: Wind and shear related variables such as wind speed and wind direction, or the dissipation of kinetic energy in the boundary layer.

- Cloud physics: Everything directly related to clouds such as the mass of various cloud particles, precipitation measures,

or the cloud size.

- Moisture field: Humidity related variables, such as dew point temperature, moisture divergence, or total humidity.

- Surface exchange: Boundary layer height and fluxes between the surface and the atmosphere such as latent and sensible heat.

For multivariate data analyses such as $k$-means cluster analysis and PCA, it is important that the underlying variables (here:

ERA5) are on the same scale and follow distributions as similar as possible. To mitigate the pronounced skewness of most of the ERA5 variables, all of them are transformed by taking square roots:

$$x_t = sign(x)\sqrt{abs(x)} \tag{1}$$

where $x$ denotes the original ERA5 variable and $x_t$ its transformation.

Moreover, to make deviations from "normal" levels comparable across variables, all variables are scaled to a mean of zero

and standard deviation of one on the scenarios without lightning.

$$x_s = \frac{(x_t - \mu)}{\sigma} \tag{2}$$

where $x_s$ denotes the scaled value. $\mu$ and $\sigma$ are the empirical mean and standard deviation based on all cell-hours in winter and in summer *without* lightning. The applied algorithm is supplied in the supplements of this paper.

### 3.3   Statistical methods

To group the 6,304 cell-hours of 35 ERA5 variables each into similar groups $k$-means clustering (MacQueen, 1967; Hartigan and Wong, 1979) is employed. Given the desired number of clusters $k$, the $k$ clusters are chosen so that the sum of squared Euclidean distances of each cell-hour to the nearest cluster mean is minimized. This minimization problem is solved iteratively using the algorithm of MacQueen (1967) with 150 different sets of starting values in order to avoid getting stuck in local minima. $k$ is set to five clusters because the sum of squared distances clearly decreases for every additional cluster until $k = 5$

but levels out for more than five clusters. Analyzing dendrograms from hierarchical clustering further support this decision.

Principal component analysis (PCA, Mardia et al. 1995) is a statistical method for dimension reduction that tries to find maximal variability within projections of the data. Each principal component (PC) is a linear combination of projected input data and is oriented perpendicular to the previous principal components. The principal components are ranked by the variance they explain so that the most variance within the data is captured by the first few principal components. Independent of the

cluster analysis, the PCA is applied to the 6,304 cell-hours of 35 ERA5 variables. The resulting first two principal components are used for visualizing the 35-dimensional data in a two-dimensional so-called biplot to facilitate interpretation. PC 1 and PC 2 are sufficient for a reasonable interpretation because they together explain about 50 % of the variance within the data, whereas the additional explained variance of PC 3 is already down to 7.6 %. The R code replicating the clustering and principal component analysis of the presented sample is provided as an online supplement along with this paper.

## 4   Results

In this section, we first present the results of the cluster analysis and the PCA, which reveals that most lightning in winter is explained by wind-field variables while most lightning in summer is explained by mass-field variables (Sect. 4.1). Then we interpret the clusters meteorologically in more detail. Wind-field thunderstorms are associated with shallow, rather warm clouds and high horizontal wind speed and shear. CAPE-thunderstorms are associated with increased values in the mass-field with

large CAPE values, high $-10\,°C$ isotherm heights and deep, cold clouds (Sect. 4.2). Finally, we look at synoptic scale processes related to the clusters and find that wind-field thunderstorms occur in the region of cyclogenesis and are characterized by strong westerly flow while CAPE-thunderstorms occur on the anticyclonic side of the jet with south-westerly flow (Sect. 4.3).

### 4.1   Cluster and principal component analysis

The statistical procedure of clustering ERA5 variables and applying a principal component analysis gives a physically in-

terpretable result. Figure 2 shows the 6,304 cell-hours of the dimension-reduced ERA5 variables, projected onto the two-dimensional space of the first two principal components (PC 1 and PC 2; axes). Each cell-hour is represented by a color-coded symbol that indicates to which of the five clusters it belongs. The five clusters are located in different parts of the span of the first two principal components. The cell-hours in the clusters symbolized by dark red triangles and dark blue circles occupy the outer reaches of the upper and lower right quadrants respectively, each covering approximately 7 % of all cell-hours. Closer to

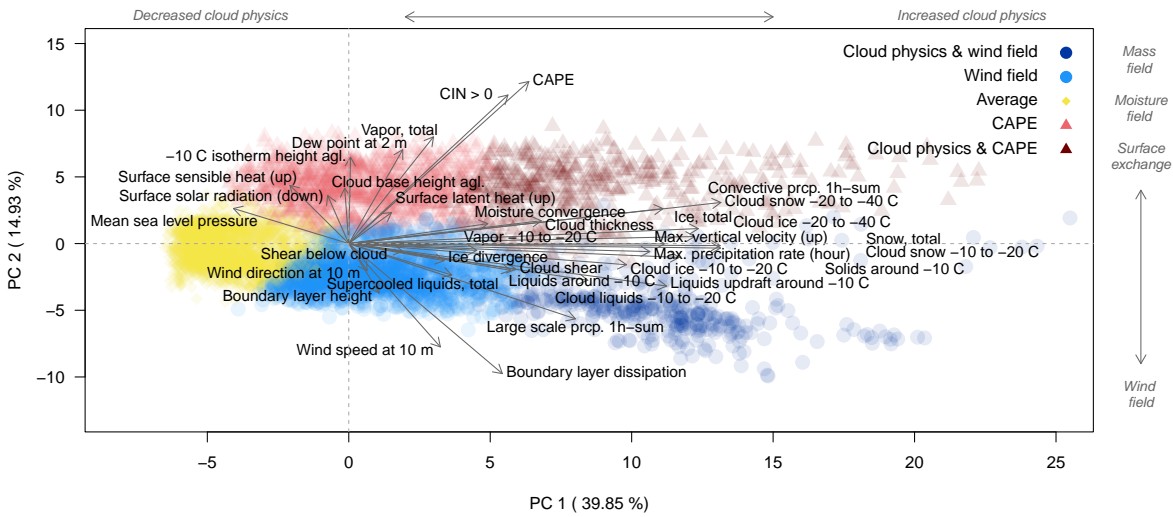

**Figure 2.** Plot of the 6,304 cell-hours separated into five clusters by $k$-means clustering (colored symbols) and then projected onto their first two principal components (PC 1 and PC 2). Labeled arrows (biplot) show the loading of each variable (35 in total), i.e. how it contributes to creating the first two principal components. The top and right axes are labeled (in italics) to indicate the dominant physical categories defined in Sect. 3.2. Note that the orientation of the arrows in the surface exchange category depend largely on how the flux direction is defined.

the origin in the upper two quadrants, the cluster symbolized by light red triangles covers approximately 17 % of the cell-hours and the cluster in the lower two quadrants with the light blue circles covers approximately 27 %. The largest cluster (41 %) depicted by yellow diamonds is closest to the origin, i.e. the values of the ERA5 variables in these cell-hours are close to average. Accordingly, we label this cluster "average". To find a possible physical meaning of the other four clusters, the so-called "loadings" from the PCA are examined.

The loadings are shown as labeled arrows in Fig. 2. Their length and direction depict how each variable contributes to creating the first two principal components. The loadings of most variables from the cloud-physics category have a large component parallel to the axis of the first principal component (PC 1). Accordingly, the upper axis in the figure is labeled as "cloud physics" (increased vs. decreased). The loadings of the variables from the other four physical categories, on the other hand, have a larger component parallel to the second principal component PC 2. The right axis in the figure is labeled

accordingly, yielding the physical meaning of the remaining four clusters.

   The light red cluster extends largely along the positive part of the second principal component that is dominated by variables of the mass-field and moisture-field categories, especially CAPE. It is accordingly named "CAPE-thunderstorm" cluster. The dark red cluster in the upper right quadrant with a large component along both PCs can thus be termed the "cloud-physics & CAPE-thunderstorm" cluster. Analogously, the light blue cluster is dominated by the wind-field category and termed "wind-

field thunderstorm" cluster, and the dark blue one "cloud-physics & wind-field thunderstorm" cluster.

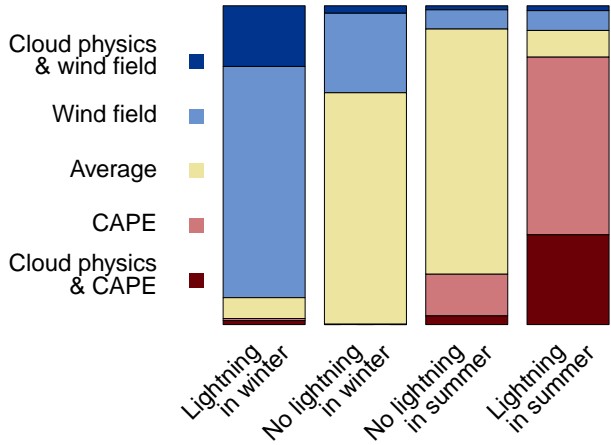

**Figure 3.** Stacked bar plot of the clusters (colors, y-axis) found in the different scenarios (bars, x-axis).

Reducing the number of clusters in the cluster analysis leads to a combined "cloud-physics" cluster ($k = 4$) and a large cluster uniting "wind-field thunderstorms" with "CAPE-thunderstorms" ($k = 3$). This stresses how well the cluster analysis differentiates between lightning and no lightning in general and points to the importance of the cloud-physics variables to distinguish between thunderstorm types.

After having discovered that the five clusters correspond to different atmospheric processes and variables, Fig. 3 shows that they also neatly fit into the four seasonal scenarios (winter vs. summer with and without lightning). The scenario of lightning in winter is dominated by the clusters termed wind-field thunderstorms (light blue), and cloud-physics & wind-field thunderstorms (dark blue); only a tiny fraction of the cloud-physics & CAPE-thunderstorm cluster contributes to it. The situation is reversed in the summer lightning scenario where the CAPE-thunderstorm cluster and the cloud-physics & CAPE-thunderstorm cluster dominate (reds). However, some events from the wind-field thunderstorm cluster also occur. The two no-lightning scenarios are dominated by the average cluster (yellow) with some contributions of the wind-field cluster in winter and of CAPE-thunderstorms in summer. Unsurprisingly, the separation between lightning and no-lightning scenarios with reanalysis variables is not completely sharp. But surprisingly clearly, the situations where wind-field variables dominate with large deviations from their average values correspond to the lightning cases in winter. In summer, on the other hand, large deviations from average in the mass field dominate the lightning cases, and only few wind-field dominated cases occur.

Extending our analysis to the full year (see supplements) reveals that spring and fall both consist of around 36 % CAPE-thunderstorms, 25 % wind-field thunderstorms, 20 % cloud-physics & CAPE-thunderstorms, and 10 % cloud-physics & wind-field thunderstorms.

## 4.2 Meteorological characterization of the clusters

Next, we zoom into the clusters and interpret the variables aggregated to them from a meteorological perspective.

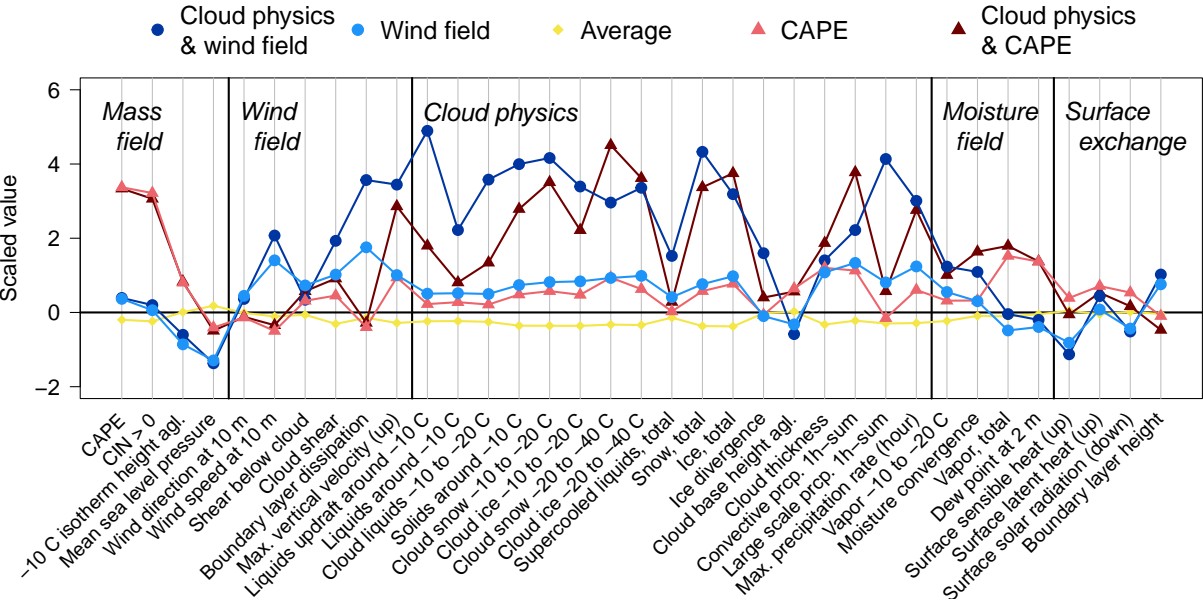

**Figure 4.** Cluster means (color-coded) of scaled ERA5 variables arranged by physical categories (italics). Variables are transformed by square root and standardized to a zero mean and a standard deviation of one on the scenarios without lightning.

Figure 4 shows the cluster means of all 35 ERA5 variables, the corresponding unscaled cluster medians are presented in Table 2. The variables are grouped by their respective physical category (mass field, wind field, cloud physics, moisture field, and surface exchange processes). Values in the "average" cluster (yellow) are close to zero, i.e. their mean. Since the average cluster contains the no-lightning situations (cf. Fig. 3), which make up the predominant state of the atmosphere, variables are expected to be in their typical range. This corroborates again that clustering reflects physical meaning. Figure 4 (cluster analysis) confirms what the loadings in Fig. 2 (PCA) already indicated: Variables with larger arrows towards a given cluster in Fig. 2 correspond to higher values for that cluster in Fig. 4.

**CAPE-thunderstorm clusters**

Figure 4 shows that indeed most mass-field variables have large deviations from their average for the events separated into the "CAPE-thunderstorm" clusters (reds). The layer crucial for the occurrence of charge separation – represented by the $-10\,°C$ isotherm – is high above the ground (median above $5\,km$, see Table 2), which is typical for summer, for which the CAPE-thunderstorm clusters prevail. Also total column water vapor (humidity) and $2\,m$ dew point from the moisture field category is increased. CAPE represents both, mass field and moisture field variables and is high only in the CAPE-thunderstorm clusters with median values of $420\,J\,kg^{-1}$. When large values of CAPE are released, tall (cumulonimbus) clouds can form and convective precipitation ensues. Accordingly, events in the CAPE-thunderstorm clusters also have high values in some variables of the other physical categories: From the cloud-physics category, the cloud size, convective precipitation, and maximum precip-

itation rate are increased. From the wind-field category, shear and vertical velocity are increased. Tall clouds are more likely to have higher shear across their depth and release of CAPE leads to larger vertical velocities. Overall, CAPE-thunderstorms are responsible for most flashes in our study region because $84\,\%$ of the lightning cell-hours in summer (JJA) are clustered as CAPE-thunderstorms. As summer is the main lightning season in our study region, we expect CAPE-thunderstorm processes to be the predominant lightning mechanism there.

**Wind-field thunderstorm clusters**

Figure 4 and Table 2 confirm that the values of wind-field variables of the cell-hours grouped into the wind-field thunderstorm clusters (blue lines) are indeed unusually large. Wind speeds, shear, and dissipation of kinetic energy in the boundary layer are all large. High shear also contributes to a larger and downward oriented sensible heat flux (from the physical category of surface fluxes). Increased mechanical mixing, in turn, leads to deep (mixed) boundary layer heights of median more than $1\,\mathrm{km}$, even with low solar energy input. As Fig. 3 shows, events in the wind-field clusters occur mostly during winter. Accordingly, the $-10\,^{\circ}\mathrm{C}$ isotherm is closer to the ground (median around $2,5\,\mathrm{km}$) and surface dew point and total column water vapor (from the moisture field category) are lower. Surface temperatures in the study region are mostly low but above freezing and in a rather narrow range (not shown) for events in the wind-field clusters. Likely, strong shear and mechanical mixing, possibly aided by the presence of clouds will prevent the build-up of nocturnal cold pools. CAPE is around $22\,\mathrm{J\,kg^{-1}}$ and therefore close to its normal value of zero. Unusually low mean sea level pressure (from the mass-field category) hints at the reason for high wind speeds and shear: mid-latitude low pressure systems and their associated strong baroclinicity, which leads to larger values of vertical shear via the thermal wind relationship.

Figure 5 presents clusterwise vertical profiles for wind speed. Events in the wind-field thunderstorm cluster (light blue) have wind speeds about twice as high as events in the CAPE-thunderstorm (light and dark red) and average (yellow) clusters, respectively. Median wind speeds for those events, where cloud-physics variables are particularly large (dark blue; discussed in more detail in the next section) are even three times as large. Within the lowest kilometer, wind speeds in the wind-field cluster (light blue) increase by more than $20\,\mathrm{m\,s^{-1}}$. Since median speeds further up to almost $4\,\mathrm{km}$ above sea level remain constant, horizontal temperature gradients in this layer must be small. Overall, this shape of the wind profile is typical of strong wintertime cyclones and their associated cold fronts. For events in the CAPE-thunderstorm clusters (reds), which occur in the warm season (cf. Fig. 3), wind shear is much lower. There, the wind speeds increase only by about $10\,\mathrm{m\,s^{-1}}$ in the lower half of the troposphere up to $5\,\mathrm{km}$. Strong summertime convection is driven by the release of CAPE with wind shear playing a secondary role in organizing this convection. Our observed values of $10\,\mathrm{m\,s^{-1}}$ difference in horizontal wind speeds between the lower and upper troposphere for CAPE-thunderstorms (reds, Fig. 5)) point to the well-known fact that most summertime thunderstorms are single cells or multicells (Markowski and Richardson, 2010). The large values of CAPE allow vertical velocities of $10-20\,\mathrm{m\,s^{-1}}$ and more within thunderstorms, exceeding the horizontal wind speeds resulting in a mainly vertical separation path of the particles. For the wind-field thunderstorms, the horizontal wind speeds in the lower troposphere are comparable or higher to the updrafts and might thus separate differently charged and differently sized cloud particles also in

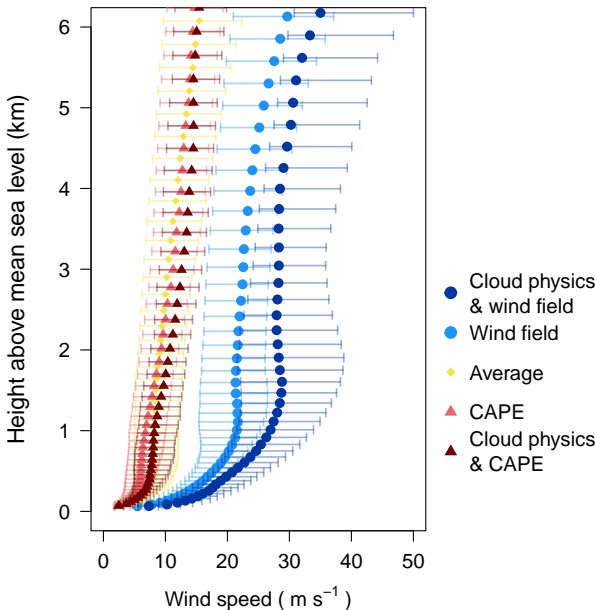

**Figure 5.** Clusterwise medians along with upper and lower quartiles of wind speed at each model level in ERA5 plotted at the mean model level height of the respective clusters.

255 the horizontal direction. This supports the hypothesis of shallow but tilted charge regions for lightning in winter (Takeuti et al., 1978; Brook et al., 1982; Williams, 2018).

**The role of cloud physics within the lightning involving clusters**

Cloud physical details are crucial for lightning to occur in general. Figure 3 shows that the "average" cluster contains most of the non-lightning events and accordingly the cloud-physics variables are close to their scaled mean of zero (Fig 4). In con-

260 trast, events in the wind-field thunderstorm (blues) and CAPE-thunderstorm (reds) clusters come with lightning (Fig. 3) and the scaled values of most of their cloud-physics variables are elevated above zero. Yet the clustering algorithm detected two groups of events with vastly elevated values of the cloud-physics variables (dark blue and dark red). Together these two groups cover 24 % of the data in the lightning involving clusters and would merge when reducing the number of clusters to $k$=4. They have much higher cloud particle concentrations compared to the other lightning involving clusters. Consequently, these are

265 events when thick clouds with large amounts of particles needed for charge separation are present in the ERA5 reanalysis. Of secondary importance are then either wind-field variables, putting these events into the "cloud-physics & wind-field" cluster, which occur in winter (cf. Fig. 3), or mass-field variables, putting them into the "cloud-physics & CAPE-thunderstorm" cluster, which occur in summer. The wintertime cloud-physics & wind-field cluster is accompanied by some vastly elevated values of wind-field variables, whereas the summertime cloud-physics & CAPE-thunderstorm cluster differs from the CAPE-

270 thunderstorm cluster only by elevated values of cloud physics, not in mass-field values. The type of precipitation that occurs for

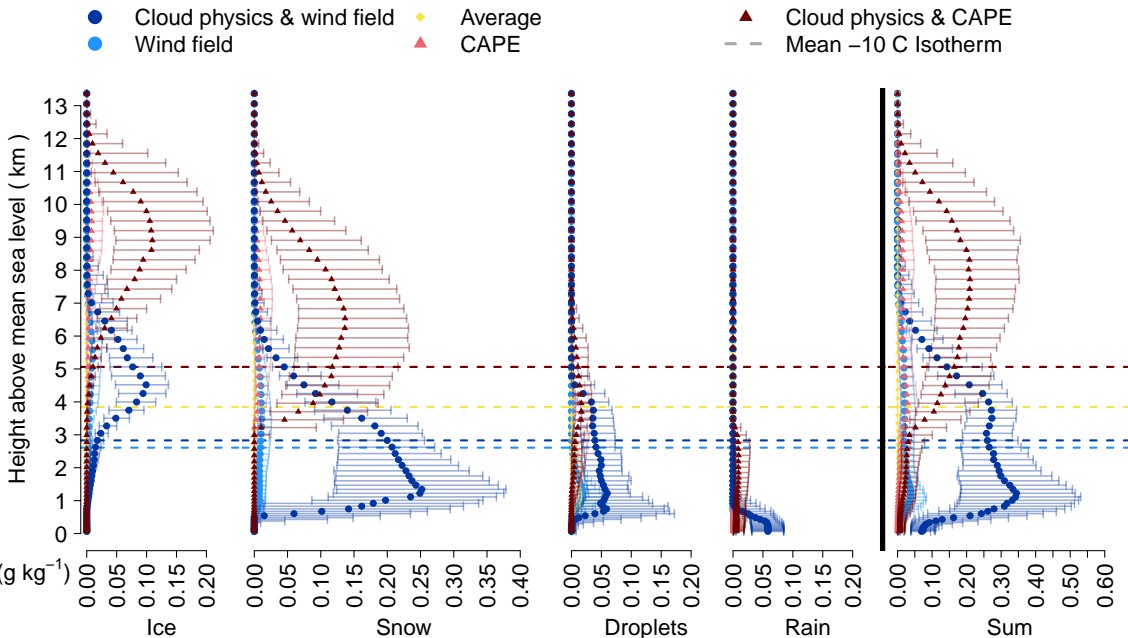

**Figure 6.** Clusterwise medians along with upper and lower quartiles of suspended particles (ice crystals, droplets) and hydrometeors (snow, rain) at each model level in ERA5 plotted at the mean model level height of the respective clusters. For each cluster the mean height of the $-10\,°C$ isotherm is included as dotted line. The last panel displays the sum of the other four and uses a different scale.

events in these cloud-physics clusters indicates again the accompanying weather types. Wintertime events in the cloud-physics & wind-field cluster come with unusually large values of large scale precipitation indicative of large scale slanted ascent in mid-latitude cyclones, whereas precipitation from convection plays a minor role. The opposite is the case for events in the summertime cloud-physics & CAPE-thunderstorm cluster. There, precipitation is mostly from convection (i.e. vertical ascent).

Some cloud-physics variables, such as the cloud size, the distribution of cloud particles relevant for charge separation, and the temperature, are better understood when looking at their vertical profiles. Figure 6 shows such profiles for suspended particles (ice crystals, droplets), hydrometeors (snow, rain), and their sums along with the mean $-10\,°C$ isotherm height for each cluster. The large difference between the clusters with enhanced cloud physics (dark blue and dark red) and their moderate counterparts (light blue and light red) is directly visible because their quartiles do not intersect over large areas.

Regarding the cloud size, Fig. 6 shows that the cloud base during events in the wind-field clusters (blues) is approximately 1 km lower than for events in the CAPE-thunderstorm clusters (reds; lowest level in the sum or droplets panel or Table 2). Cloud tops in the wind-field clusters are approximately 5 km shallower, having cloud top heights at around 7 km versus 12 km in CAPE-thunderstorm clusters (highest levels in the sum or ice panel). Put differently, considering that wind-field thunderstorm events occur in winter and CAPE-thunderstorm events in summer, thunderstorm clouds in winter are lower-based and

considerably shallower than in summer. This has a somewhat surprising consequence on the temperatures of these clouds. Looking at the cloud mass (sum of all cloud particles) below and above the $-10\,°C$ isotherm (dashed lines) of wind-field thunderstorm clouds (blues), the larger part (factor 1.7 without and factor 2.3 with cloud physics) is warmer than $-10\,°C$. CAPE-thunderstorm clouds (reds) have similar or larger cloud particle concentrations (factor 1 without and factor 2.9 with cloud physics) in regions that are colder than $-10\,°C$ resulting in rather cold clouds. Hence, during lightning in winter clouds are – integrated over their depth – overall warmer than summer clouds.

The shape of the vertical cloud particle distribution is consistent with the possibility of charge separation to have occurred (panels 1–4). Both the formation of a graupel dipole and a snow dipole, respectively, require a spatial separation of light ice crystals and heavier solid hydrometeors after their charge transferring collisions [1] in the presence of supercooled liquid water. And indeed, for events in the wind-field thunderstorm and CAPE-thunderstorm clusters ice crystal maxima (ice panel) are several kilometers above the solid hydrometeor maxima (snow panel) and the zone of cloud liquids (droplets panel) include the $-10\,°C$ isotherm. Events in the no-lightning average cluster (yellow) either have no or only shallow clouds, which consist mostly of suspended droplets so that no charge separation is possible.

## 4.3 Weather patterns

The clusters found by the cluster analysis are not only associated with typical variables and seasons but also with typical weather patterns. Figure 7 shows median weather patterns for the three largest clusters. The clusters with enhanced cloud physics are not shown since weather patterns are similar to those of their moderate counterparts. Wind speed (color) and anomalies of geopotential height (black lines) at 300 hPa are plotted along with anomalies of temperature (red dotted lines) at 700 hPa.

Events grouped into the wind-field thunderstorm cluster (Fig. 7 a) have a strong inflow from west-northwest towards the study region in northern Germany, as the tightly packed isohypses (black lines) show. The study region is located in the left exit region and at the cold and cyclonic side of the jet, where cyclogenesis and ascent take place as can be shown using ageostrophic circulation reasoning (e.g. Martin, 2006). At 700 hPa, a substantial horizontal NE–SW temperature gradient becomes apparent (approximately $8\,°C$ per $1{,}000$ km). Lightning events in the CAPE-thunderstorm clusters (Fig. 7 b) predominantly originate in south-west weather patterns. The study region is situated at the warm and anticyclonic side of the jet, prevalently in the warm sector of the frontal systems. Ageostrophic circulations favor large scale descent. However, advection of warm and moist air from the Mediterranean Sea potentially increases CAPE with convection ensuing when it is triggered and released. Events in the average cluster (Fig. 7 c) mostly lack lightning. While they are a composite of various weather patterns, the zonal pattern of the isohypses reflects the predominance of westerly flow as a result of the north-south oriented temperature gradient typical of a mid-latitude region.

---

[1]Graupel and snow are not distinguished in the ERA5 reanalysis, which has only one summarizing variable of solid hydrometeors.

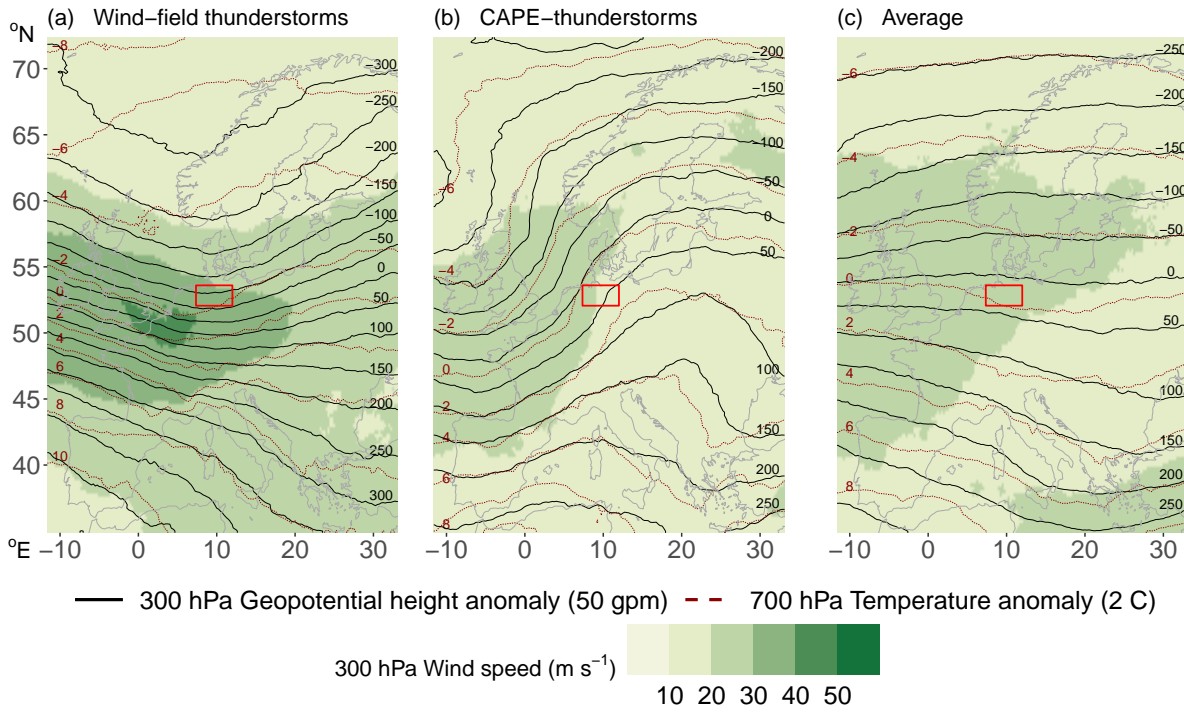

**Figure 7.** Median weather charts for the clusters in the observational region (red rectangle) showing wind speed (colors) and anomalies of geopotential height relative to the mean (solid black lines) at 300 hPa, and temperature anomalies at 700 hPa (dotted red lines). Number of charts composed for each cluster: wind field: 1,729, mass field: 1,096, and average: 2,591.

## 5   Discussion

Rather than taking the common approach of looking at differences between thunderstorms in winter and summer, we have taken a data-driven approach. Starting with a large set of variables that are potentially important for the formation of lightning (e.g., Vogel et al., 2016; Kolendowicz et al., 2017), and putting them through a clustering and principal component analysis yielded four physically meaningful clusters that distinguish different types of thunderstorms. In the first type (cf. Fig. 4), variables in the mass-field category such as CAPE, CIN, or the height of the $-10\,°\mathrm{C}$ isotherm deviate strongly from their average values ("CAPE-thunderstorms"). In the second type, variables in the wind-field category such as shear within the cloud, 10 m wind speed, or boundary layer dissipation deviate strongly ("wind-field thunderstorms"). The other two types are variants of the previous two but have additionally pronounced deviations in variables within the cloud-physics category such as the mass of solid cloud particles, or precipitation amounts ("cloud-physics & wind-field thunderstorms"; "cloud-physics & CAPE-thunderstorms").

The clear distinction between thunderstorm types characterized by high values in either the wind field or the mass field highlights that thunderstorms should not be conflated with strong convection. Strong moist convection depends upon high vertical velocity and deep clouds, which requires the presence of CAPE and a trigger to release it. Only CAPE-thunderstorms

fulfill these requirements, while CAPE in wind-field thunderstorms is basically zero. However, the defining characteristic of a thunderstorm is thunder caused by lightning (WMO, 1992) and lightning occurs when differently charged regions in a cloud equalize. Those charged regions are thought to form when different cloud particles collide and are subsequently spatially separated by differential terminal velocities (e.g., Williams, 2018). In CAPE-thunderstorms, vertical velocities are usually large ($10-50\,\mathrm{m\,s^{-1}}$) when CAPE is released but in wind-field thunderstorms, CAPE is too small ($\sim 22\,\mathrm{J\,kg^{-1}}$) to explain the necessary vertical motions. Instead, it seems that high horizontal wind speeds and large vertical shear of the horizontal wind cause the charge separation (cf. Fig. 5 and Table 2)). Separation of the charge regions is then no longer predominantly in the vertical but strongly tilted – known as "tilted charge hypothesis" (Takeuti et al., 1978; Brook et al., 1982; Engholm et al., 1990; Williams, 2018; Takahashi et al., 2019; Wang et al., 2021). These tilted charge regions were first observed in Japan during winter with high, strongly sheared horizontal wind speeds (Takeuti et al., 1978; Brook et al., 1982) and have since been observed in (mesoscale convective) storms in winter and summer (Brook et al., 1982; Engholm et al., 1990; Levin et al., 1996; Dotzek et al., 2005; Liu et al., 2011; Takahashi et al., 2019). The discussion is often accompanied by an analysis of increased positive lightning discharges in winter (Takeuti et al., 1978; Brook et al., 1982; Takagi et al., 1986; Takahashi et al., 2019; Wang et al., 2021). Observations of longer lightning channels in high-wind conditions (López et al., 2017; Yoshida et al., 2018) further support the tilted charge hypothesis.

Whether a wind-field thunderstorm or CAPE-thunderstorm occurs depends on the larger-scale synoptic environment. In the northern Germany study region, the prevalence of these environments strongly varies seasonally. Weather patterns with unusually large values in wind-field related variables (cf. Fig. 7 a) dominate in winter. Accordingly, the wind-field thunderstorms occur mostly in the cold season. Similar weather patterns as in Fig. 7 a with strong, mostly zonal flow, and high wind speeds are found in wintertime studies of thunderstorm days in central-eastern Europe (Kolendowicz et al., 2017) and derechos (high-impact convective wind events) in winter in Germany (Gatzen et al., 2020). Due to the stronger horizontal temperature gradients in mid-latitudinal winter, higher wind speeds and thus wind-field thunderstorms also occur in other continents, e.g., USA and Japan. For the USA, Bentley et al. (2019) have evidence that lightning in winter is often associated with the development and progression of mid-latitude cyclones and that the synoptic weather systems are more important than insolation. Our results in Fig. 7 a also locate wind-field thunderstorms into the left exit region of the jet, where cyclogenesis typically occurs (e.g., Martin, 2006). Sometimes lightning in winter is referred to as high-shear low-CAPE (HSLC) storms (Johns et al., 1993; Sherburn and Parker, 2014). However, thresholds of $500\,\mathrm{J\,kg^{-1}}$ to define "low CAPE" constitute high CAPE in our target region where wind-field thunderstorms have median values of $22\,\mathrm{J\,kg^{-1}}$ for CAPE and could thus analogously be named "high-shear no-CAPE" events.

Large-scale weather patterns leading to CAPE-thunderstorms, characterized by large CAPE values (median $415\,\mathrm{J\,kg^{-1}}$) and increased heights of the $-10\,^{\circ}\mathrm{C}$ isotherm (median $5,170\,\mathrm{m}$) dominate in the warm season in our study region. The preferred weather pattern of southwesterly flow (Fig. 7 b) was also found to be important for summertime lightning in the larger area of central Europe (Kaltenböck et al., 2009; Westermayer et al., 2016; Kolendowicz et al., 2017) and accounts for the majority lightning activity in Europe. CAPE-thunderstorms are well described in the literature and often taken to be synonymous with

thunderstorms in general (e.g., Williams et al., 2005; Kaltenböck et al., 2009; Mora et al., 2015; Stolz et al., 2017; Kolendowicz et al., 2017; Dewan et al., 2018; Etten-Bohm et al., 2021).

The statistical approach of clustering and principal component analysis found two more clusters that are variants of the wind-field thunderstorm type and CAPE-thunderstorm type and vary seasonally in the same way. For them, cloud-physics variables strongly deviate from average conditions. They point to the need for including cloud physics for the indirect diagnosis of thunderstorms from atmospheric proxy variables since cloud physics is essential for electrification.

The study area was deliberately limited to a topographically uniform region (northern Germany) to reduce the complexity
of the problem. The data-driven approach used here should easily transfer to other regions. When larger, non-homogeneous regions are studied, the data scaling techniques will have to be extended to be able to deal with spatially varying means and anomalies.

Using a lightning location system (LLS) to detect lightning misses a particular type of upward lightning, which consists of a continuous current only. Such lightning can currently only be detected at very few specially instrumented towers. While
it is rare in absolute numbers and affects only tall structures ($100+$ m, it might contribute up to half of lightning activity in winter at such locations (Diendorfer et al., 2015). Preliminary results indicate that these lightning events occur in wind-field thunderstorms, corroborating the findings of this study. (Stucke et al., 2022).

Our results show that in order to distinguish physically different thunderstorm types atmospheric variables describing wind field, mass field, and cloud physics must be included (cf. Figs. 4 and 3). Identifying thunderstorms and lightning from single
or just a few atmospheric proxy variables is inaccurate. Using only CAPE (or related) variables will even completely miss the wind-field thunderstorm class where different physical processes are at work.

## 6    Conclusions

In most mid-latitude regions, lightning in winter contributes only a few percent to the annual number of flashes. In our study region in northern Germany, there is approximately 70 times more lightning in summer than in winter. We investigated whether
the same atmospheric conditions as for summertime thunderstorms were at play in winter but only occurred much less frequently and less pronounced or whether winter thunderstorms were physically different.

Following a data-driven approach, we used 35 atmospheric variables from the ERA5 reanalysis belonging to five meteorological categories (mass field, wind field, cloud physics, moisture field, and surface exchange) and fed them independent of each other into a clustering and a principal component algorithm. These hourly data are linked to observations with and without
lightning in winter (DJF) and summer (JJA) and the variables have shown to be potentially relevant for lightning.

The statistical analysis returned four clusters (thunderstorm types), that have the same physical interpretation with respect to their cluster means and, independent from it, their loadings. The two main lightning types consist of events for which ERA5 variables in either the wind-field (wind-field thunderstorms) or the mass-field (CAPE-thunderstorms) category strongly deviate from their means. The other two types are variants of the wind-field thunderstorm and CAPE-thunderstorm respectively, for
which additionally the cloud-physics variables strongly deviate from their mean values. Our study region is struck by lightning

from wind-field thunderstorms predominantly (88 %) in the cold season, whereas CAPE-thunderstorm lightning occurs only in the warm season (98 %).

Differently-charged layers in the atmosphere are thought to come about by the collision of different types of cloud particles and hydrometeors such as ice crystals and graupel during which charge is transferred, followed by a subsequent size-dependent separation. The required terminal velocities in CAPE-thunderstorms originate from strong vertical velocities when substantial amounts of CAPE are released. Median values of CAPE in CAPE-thunderstorms in our study region is $415\,\mathrm{J\,kg^{-1}}$. For wind-field thunderstorms, the strong velocities occur mostly horizontally but with a strong vertical shear so that the charge separation happens along a slanted path.

Wind-field thunderstorms are characterized by horizontal wind speeds that approximately triple in the lowest kilometer (Fig. 5) to reach median values of more than $20\,\mathrm{m\,s^{-1}}$ and even more than $27\,\mathrm{m\,s^{-1}}$ for the variant with pronounced cloud-physics variables. Consequently, dissipation of kinetic energy in the boundary layer and boundary layer height are also increased. Synoptically, wind-field thunderstorms occur in the left exit region at the cold and cyclonic side of the jet with inflow from west-northwest. It is the region of cyclogenesis, strong updrafts, and large scale precipitation. These larger-scale patterns occur mostly in winter. Clouds are shallow and close to the ground. Especially in the thunderstorm types with enhanced cloud physics, most parts of the clouds are warmer than $-10\,^{\circ}\mathrm{C}$ and, integrated over their depth, wind-field thunderstorm clouds are warmer than CAPE-thunderstorm clouds. This results in a larger fraction of cloud droplets, warmer snow, and shallow regions consisting only of hydrometeors. The wind-field thunderstorm type with increased cloud-physics variables stands out by even larger deviations in the previously mentioned variables and occurs in similar weather patterns.

CAPE-thunderstorms have large CAPE values and convective inhibition (CIN) present and are further characterized by deep, cold clouds with a dominating region consisting of suspended ice particles and solid hydrometeors. They take place in summer. Synoptically, CAPE-thunderstorms in northern Germany occur in south-westerly flow at the anticyclonic side of the jet. Usually, warm and moist air is advected from the Mediterranean Sea. The variant of CAPE-thunderstorms with much higher values in the cloud-physics variables occur in similar weather patterns and with similar mass-field values as the CAPE-thunderstorm type. However, the clouds are deeper and have larger amounts of cloud particles accompanied by strong updrafts, and large precipitation amounts.

In summary, the data-driven approach yielded physically different types of thunderstorms, for which the defining larger-scale flow situations also vary seasonally. Winter lightning is therefore not just a weaker and rarer sibling of summer lightning but driven by wind-field variables instead of mass-field variables.

*Code and data availability.*  This paper provides an online supplement (Morgenstern et al., 2022) consisting of a precise variable description, the data of the representative sample presented here, an R-script that reproduces the core analysis and Figs. 2 - 4, and the results from an analog analysis covering also the intermediate seasons spring and fall. This supplement is available at https://zenodo.org/record/5851700.

ERA5 data are freely available at the Copernicus Climate Change Service (C3S) Climate Data Store (Hersbach et al., 2020). The results contain modified Copernicus Climate Change Service information 2020. Neither the European Commission nor ECMWF is respon-

sible for any use that may be made of the Copernicus information or data it contains. EUCLID data are available on request from ALDIS

(aldis@ove.at) or Siemens BLIDS – fees may be charged.

Calculations are performed using R (R Core Team, 2021), Python 3 (Van Rossum and Drake, 2009), and cdo (Schulzweida, 2019). Specifically the following packages: ncdf4 (Pierce, 2019), simple features (Pebesma, 2018), stars (Pebesma, 2020), rnaturalearth (South, 2017), and xarray (Hoyer and Hamman, 2017). We are using the netCDF4 data format (Unidata, 2020).

*Author contributions.*    Deborah Morgenstern did the investigation, wrote software, visualized the results, and wrote the paper. Isabell Stucke,

Thorsten Simon, and Deborah Morgenstern performed the data curation, built the data set, and derived variables based on ERA5 data. Thorsten Simon contributed coding concepts. Georg J. Mayr provided support for the meteorological analysis, data organization, and funding acquisition. Achim Zeileis supervised the formal analysis and interpretation of the statistical methods. Achim Zeileis, Georg J. Mayr, and Thorsten Simon are the project administrators and supervisors. All authors contributed to the conceptualization of this paper, discussed the methodology, evaluated the results, and commented on the paper.

*Competing interests.*    The authors declare that they have no conflict of interest.

*Acknowledgements.*    We acknowledge the funding of this work by the Austrian Research Promotion Agency (FFG), project no. 872656 and Austrian Science Fund (FWF) grant no. P 31836. We thank Johannes Horak for his code to calculate the geopotential. Furthermore, we are grateful to Gerhard Diendorfer and Wolfgang Schulz from ALDIS for data support and discussions about lightning physics and to Siemens BLIDS for providing EUCLID data. Finally, we thank the editor and two anonymous reviewers for their valuable comments.

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

**Table 1.** ERA5 variables used in this study. Variables derived from other ERA5 variables are marked by an asterisk (*).

| Name | Unit | Category |
|---|---|---|
| Boundary layer dissipation | $J\,m^{-2}$ | wind field |
| Boundary layer height | m | surface exchange |
| CAPE | $J\,kg^{-1}$ | mass field |
| CIN > 0 | binary | mass field |
| Cloud base height agl. | m a.g.l. | cloud physics |
| Cloud ice $-10$ to $-20\,°C$ * | $kg\,m^{-2}$ | cloud physics |
| Cloud ice $-20$ to $-40\,°C$ * | $kg\,m^{-2}$ | cloud physics |
| Cloud liquids $-10$ to $-20\,°C$ * | $kg\,m^{-2}$ | cloud physics |
| Cloud shear * | $m\,s^{-1}$ | wind field |
| Cloud thickness * | m | cloud physics |
| Cloud snow $-10$ to $-20\,°C$ * | $kg\,m^{-2}$ | cloud physics |
| Cloud snow $-20$ to $-40\,°C$ * | $kg\,m^{-2}$ | cloud physics |
| Convective prcp. 1h-sum | m | cloud physics |
| Dew point at 2 m | K | moisture field |
| Ice divergence | $kg\,m^{-2}\,s^{-1}$ | cloud physics |
| Ice, total | $kg\,m^{-2}$ | cloud physics |
| Large scale prcp. 1h-sum | m | cloud physics |
| Liquids around $-10\,°C$ * | $kg\,m^{-2}$ | cloud physics |
| Liquids updraft around $-10\,°C$ * | $kg\,Pa\,s^{-1}$ | cloud physics |
| Max. precipitation rate (hour) | $kg\,m^{-2}\,s^{-1}$ | cloud physics |
| Max. vertical velocity (up) * | $Pa\,s^{-1}$ | wind field |
| Mean sea level pressure | Pa | mass field |
| Moisture convergence | $kg\,m^{-2}\,s^{-1}$ | moisture field |
| Shear below cloud * | $m\,s^{-1}$ | wind field |
| Snow, total | $kg\,m^{-2}$ | cloud physics |
| Solids around $-10\,°C$ * | $kg\,m^{-2}$ | cloud physics |
| Surface latent heat (up) | $J\,m^{-2}$ | surface exchange |
| Surface sensible heat (up) | $J\,m^{-2}$ | surface exchange |
| Surface solar radiation (down) | $J\,m^{-2}$ | surface exchange |
| Supercooled liquids, total | $kg\,m^{-2}$ | cloud physics |
| Vapor $-10$ to $-20\,°C$ * | $kg\,m^{-2}$ | moisture field |
| Vapor, total | $kg\,m^{-2}$ | moisture field |
| Wind direction at 10 m * | ° | wind field |
| Wind speed at 10 m * | $m\,s^{-1}$ | wind field |
| $-10\,°C$ isotherm height agl. * | m a.g.l. | mass field |

agl = above ground level, prcp = precipitation

**Table 2.** Cluster medians

| Variable | Unit | Cloud physics & wind field | Wind field | Average | CAPE | Cloud physics & CAPE |
|---|---|---|---|---|---|---|
| CAPE | $J\,kg^{-1}$ | 22 | 22 | 1 | 415 | 425 |
| CIN $> 0$ | binary | 0 | 0 | 0 | 1 | 1 |
| $-10\,°C$ isotherm height | m a.g.l. | 2,629 | 2,234 | 4,160 | 5,171 | 5,244 |
| Mean sea level pressure | hPa | 1,003.9 | 1,003.4 | 1,016.8 | 1,011.4 | 1,010.4 |
| Wind direction at 10 m | ° | 249 | 250 | 227 | 214 | 216 |
| Wind speed at 10 m | $m\,s^{-1}$ | 8.8 | 6.6 | 3.4 | 2.7 | 2.9 |
| Shear below cloud | $m\,s^{-1}$ | 8.3 | 10.0 | 4.9 | 6.0 | 8.2 |
| Cloud shear | $m\,s^{-1}$ | 29.7 | 17.6 | 3.8 | 10.3 | 15.2 |
| Boundary layer dissipation | $W\,m^{-2}$ | 34.4 | 14.5 | 1.9 | 1.4 | 1.7 |
| Max. vertical velocity (up) | $Pa\,s^{-1}$ | 1.33 | 0.41 | 0.14 | 0.36 | 1.00 |
| Liquids updraft around $-10\,°C$ | $g\,Pa\,s^{-1}$ | 22.77 | 0.26 | 0 | 0.01 | 2.83 |
| Liquids around $-10\,°C$ | $g\,m^{-2}$ | 24.56 | 2.68 | 0 | 1.63 | 5.12 |
| Cloud liquids $-10$ to $-20\,°C$ | $g\,m^{-2}$ | 51.2 | 1.42 | 0 | 0.73 | 6.67 |
| Solids around $-10\,°C$ | $g\,m^{-2}$ | 128.99 | 8.32 | 0 | 5.81 | 66.7 |
| Cloud snow $-10$ to $-20\,°C$ | $g\,m^{-2}$ | 216.66 | 13.30 | 0 | 9.30 | 145.21 |
| Cloud ice $-10$ to $-20\,°C$ | $g\,m^{-2}$ | 67.15 | 5.75 | 0 | 3.15 | 27.37 |
| Cloud snow $-20$ to $-40\,°C$ | $g\,m^{-2}$ | 82.61 | 14.35 | 0.01 | 11.63 | 158.57 |
| Cloud ice $-20$ to $-40\,°C$ | $g\,m^{-2}$ | 143.48 | 20.05 | 0.04 | 9.59 | 147.19 |
| Supercooled liquids, total | $g\,m^{-2}$ | 103.30 | 31.86 | 6.23 | 17.14 | 27.54 |
| Snow, total | $g\,m^{-2}$ | 851.8 | 55.2 | 0.8 | 43.7 | 512.3 |
| Ice, total | $g\,m^{-2}$ | 245.2 | 46.4 | 2.5 | 36.6 | 285.8 |
| Ice divergence | $g\,m^{-2}\,h^{-1}$ | 72.8 | -3.0 | -0.2 | -1.2 | 16.4 |
| Cloud base height | m a.g.l. | 282 | 450 | 672 | 1,283 | 1,362 |
| Cloud thickness | m | 7,125 | 6,440 | 1,234 | 8,410 | 10,645 |
| Convective prcp. 1h-sum | mm | 0.32 | 0.09 | 0 | 0.03 | 0.47 |
| Large scale prcp. 1h-sum | mm | 0.69 | 0.04 | 0 | 0 | 0.02 |
| Max. precipitation rate (hour) | $mm\,h^{-1}$ | 1.59 | 0.30 | 0 | 0.03 | 1.02 |
| Vapor $-10$ to $-20\,°C$ | $kg\,m^{-2}$ | 2.13 | 1.51 | 0.96 | 1.38 | 2.00 |
| Moisture convergence | $kg\,m^{-2}\,h^{-1}$ | 0.88 | 0.19 | -0.04 | 0.22 | 1.92 |
| Vapor, total | $kg\,m^{-2}$ | 13.9 | 10.4 | 15.5 | 33.7 | 38.1 |
| Dew point at 2 m | K | 276.8 | 275.7 | 280.4 | 289.8 | 289.9 |
| Surface sensible heat (up) | $W\,m^{-2}$ | -63 | -44 | -4 | 15 | -8 |
| Surface latent heat (up) | $W\,m^{-2}$ | 83 | 40 | 18 | 107 | 69 |
| Surface solar radiation (down) | $W\,m^{-2}$ | 18 | 1 | 41 | 207 | 84 |
| Boundary layer height | m | 1,433 | 1,143 | 595 | 556 | 429 |

agl = above ground level, prcp = precipitation