# Peer review of "Differentiating lightning in winter and summer with characteristics of wind field and mass field"

_Weather and Climate Dynamics, 2021_

## Author Response (AR1)

Johannes Dahl Co-Editor

Referee 1 and 2

Date January 14, 2022

Revision of WCD-2021-68

Dear Johannes Dahl and Referees,

please find attached the revised version of our manuscript "Differentiating lightning in winter and summer with characteristics of wind field and mass field" (WCD-2021-68).

Thank you very much for the constructive and helpful feedback from you, the associate editor, and the referees. Based on this feedback, the manuscript has been revised. We feel that this substantially improved our contribution. The most important changes are:

- *Renaming*. The former mass-field cluster is renamed into CAPE-thunderstorms cluster.
- Intermediate seasons: spring and fall. We extended our analysis to the intermediate seasons spring and fall and provide this analog analysis with the supplements. Within the manuscript, we refer more to the role of seasons.
- Limitations.

A section describing the limitations of our study is now included. Throughout the manuscript, we state more clearly where our results are valid.

• Methods.

The methods chapter is improved by several clarifications and more detailed information. In particular, the applied transformation and scaling are now described more explicitly in terms of mathematical formulas.

All changes and additions are explained in much more detail in the point-to-point reply on the next pages.

Best regards,

). Morgenster

The authors

**Referee 1**

I thank the authors for their novel empirical investigation of thunderstorm conditions, and for their work to utilize the multi-parameter empirical information to reason about the meteorological dynamics. I think the manuscript is suitable for publication with minor revisions, and will be a helpful practical forecasting aid while supporting clear, physically-based forecast reasoning.

Thank you.

I wanted to comment on the choice of words for the two categories of lightning identified in the study.

"Wind" and "mass/moisture" could be made even more specific and descriptive. The word choice here matters because it conditions how the reader perceives the connection to what actually drives the cloud physics processes that can result in electrification. To this point, the authors note that "wind field" is, specifically, the synoptic-scale thermal wind (line 200), and implies quasi-geostrophic dynamics driving clouds formation. Likewise, mass/moisture is really the specific thermodynamics associated with conditional instability and upright moist convection. These more specific ideas are strongly implied by the authors' reasoning, and so for this reason I encourage the authors to consider adopting more specific terms.

A good name concisely describes the characteristics of a phenomenon without being wordy or introducing new abbreviations. We argue that no single variables but the interplay of variables within a certain group are important to distinguish thunderstorms associated with the wind field, the mass/moisture field, or enhanced cloud physics. Further, a name should be easy to remember. We understand, that the term mass field might not be familiar to some readers. Considering your thoughts we came up with the term CAPE-thunderstorm instead of mass-field lightning. CAPE already includes mass-field variables and moisture-field variables and is a term that is easy to remember. Regarding wind-field lightning, we found no other term describing better the characteristics of this thunderstorm type. We have thought about "shear-thunderstorms", but this reduces the complexity too much because it omits wind speed and updrafts. Therefore we have kept the term wind-field thunderstorm but we are ready to rename it when somebody comes up with a suitable suggestion.

ACTION 1: Renaming mass-field thunderstorm into CAPE-thunderstorm.

ACTION 2: Arguing for the term CAPE-thunderstorm by pointing more towards the also increased moisture-field variables in these clusters:

Line 181: » The light red cluster extends largely along the positive part of the second principal component that is dominated by variables of the mass-field and moisture-field categories, especially CAPE. It is accordingly named "CAPE-thunderstorm" cluster. «

Line 217: » Also total column water vapor (humidity) and 2 m dew point from the moisture field category is increased.  $\ensuremath{^{\circ}}$

**1. If the authors reduce the number of clusters to k=3, do the dark and light, red and blue markers combine more with each other than they do with the yellow markers?**

The decision to use k = 5 is based on the lowest sum of squared distances and the analysis of the dendrogram from hierarchical clustering. In our opinion, k = 5 gives the best solution. However, k = 4 performs not too bad and k = 3 is in some samples reasonable. To answer your question, we provide two more versions of the paper's Fig. 2. When using k = 4 (Fig. 1), the clusters characterized by "cloud physics" merge (dark red and dark blue). With k = 3 (Fig 2) the wind-field thunderstorm cluster (light blue) and CAPE-thunderstorm cluster (light red) merge additionally. This stresses how much the cloud-physics variables contribute to the thunderstorm type and how well the cluster analysis differentiates between lightning and no lightning in general. All configurations differentiate similarly sharp between the average cluster and the others. The average cluster always contains some observations where lightning occurred and vice versa.

Figure 1: PCA-plot (Fig. 2) with color-coding based on k-means clustering with k = 4.

PCA, Figure 02 with k = 3

Figure 2: PCA-plot (Fig. 2) with color-coding based on k-means clustering with k = 3.

ACTION 1: A short discussion on various choices for *k* is included:

Line 186: » Reducing the number of clusters in the cluster analysis would lead to a combined "cloud-physics" cluster (k = 4) and a large cluster uniting "wind-field thunderstorms" with "CAPE-thunderstorms" (k = 3). This stresses how well the cluster analysis differentiates between lightning and no lightning in general and points to the importance of the cloud physics variables to distinguish between thunderstorm types. «

ACTION 2: The similarity of the clusters with cloud physics is pronounced more, as these clusters would merge first:

Line 162: » Together these two groups cover 24 % of the data in the lightning involving clusters and would merge when reducing the numb